# Aicardi Syndrome Is a Genetically Heterogeneous Disorder

**DOI:** 10.3390/genes14081565

**Published:** 2023-07-31

**Authors:** Thuong T. Ha, Rosemary Burgess, Morgan Newman, Ching Moey, Simone A. Mandelstam, Alison E. Gardner, Atma M. Ivancevic, Duyen Pham, Raman Kumar, Nicholas Smith, Chirag Patel, Stephen Malone, Monique M. Ryan, Sophie Calvert, Clare L. van Eyk, Michael Lardelli, Samuel F. Berkovic, Richard J. Leventer, Linda J. Richards, Ingrid E. Scheffer, Jozef Gecz, Mark A. Corbett

**Affiliations:** 1School of Biological Sciences, Faculty of Science, University of Adelaide, Adelaide, SA 5005, Australia; 2Department of Genetics and Molecular Pathology, Centre for Cancer Biology, An Alliance between SA Pathology and the University of South Australia, Adelaide, SA 5000, Australia; 3Epilepsy Research Centre, Faculty of Medicine, Dentistry and Health Sciences, University of Melbourne, Austin Health, Heidelberg, VIC 3084, Australias.berkovic@unimelb.edu.au (S.F.B.); scheffer@unimelb.edu.au (I.E.S.); 4Alzheimer’s Disease Genetics Laboratory, School of Biological Sciences, Faculty of Science, University of Adelaide, Adelaide, SA 5005, Australiamichael.lardelli@adelaide.edu.au (M.L.); 5The Queensland Brain Institute, The School of Biomedical Sciences, Faculty of Medicine, The University of Queensland, Brisbane, QLD 4000, Australia; 6Department of Paediatrics, Faculty of Medicine, Dentistry and Health Sciences, University of Melbourne, Parkville, VIC 3052, Australia; 7Department of Medical Imaging, The Royal Children’s Hospital, Melbourne, VIC 3052, Australia; 8Adelaide Medical School and Robinson Research Institute, Faculty of Health and Medical Sciences, University of Adelaide, Adelaide, SA 5005, Australiamark.corbett@adelaide.edu.au (M.A.C.); 9Department of Molecular, Cellular, and Developmental Biology, College of Arts and Sciences, University of Colorado, Boulder, CO 80309, USA; 10Department of Neurology, Women’s and Children’s Hospital, North Adelaide, SA 5006, Australia; 11Genetic Health Queensland, Royal Brisbane and Women’s Hospital, Herston, QLD 4029, Australia; 12Queensland Children’s Hospital, South Brisbane, QLD 4101, Australia; 13Department of Neurology, The Royal Children’s Hospital, Parkville, VIC 3052, Australia; 14Murdoch Children’s Research Institute, Parkville, VIC 3052, Australia; 15Department of Neurosciences, Queensland Children’s Hospital, South Brisbane, QLD 4101, Australia; sophie.calvert@health.qld.gov.au; 16Department of Neuroscience, School of Medicine, Washington University, St Louis, MO 63110, USA; 17Florey Institute of Neuroscience and Mental Health, Parkville, VIC 3052, Australia; 18South Australian Health and Medical Research Institute, Adelaide, SA 5000, Australia

**Keywords:** X-linked, sex bias, DNA sequencing, developmental epileptic encephalopathy, wnt signalling, DNA repair

## Abstract

Aicardi Syndrome (AIC) is a rare neurodevelopmental disorder recognized by the classical triad of agenesis of the corpus callosum, chorioretinal lacunae and infantile epileptic spasms syndrome. The diagnostic criteria of AIC were revised in 2005 to include additional phenotypes that are frequently observed in this patient group. AIC has been traditionally considered as X-linked and male lethal because it almost exclusively affects females. Despite numerous genetic and genomic investigations on AIC, a unifying X-linked cause has not been identified. Here, we performed exome and genome sequencing of 10 females with AIC or suspected AIC based on current criteria. We identified a unique de novo variant, each in different genes: *KMT2B*, *SLF1*, *SMARCB1*, *SZT2* and *WNT8B*, in five of these females. Notably, genomic analyses of coding and non-coding single nucleotide variants, short tandem repeats and structural variation highlighted a distinct lack of X-linked candidate genes. We assessed the likely pathogenicity of our candidate autosomal variants using the TOPflash assay for WNT8B and morpholino knockdown in zebrafish (*Danio rerio*) embryos for other candidates. We show expression of *Wnt8b* and *Slf1* are restricted to clinically relevant cortical tissues during mouse development. Our findings suggest that AIC is genetically heterogeneous with implicated genes converging on molecular pathways central to cortical development.

## 1. Introduction

Aicardi Syndrome (OMIM: 304050; AIC) is estimated to affect 4000 individuals worldwide [1,2]. The diagnostic criteria for the disorder encompass a recognizable spectrum of malformations of cortical development including agenesis of the corpus callosum (ACC), intracranial cysts, polymicrogyria, nodular heterotopia and basal ganglia dysmorphism, early onset of infantile epileptic spasms and/or focal seizures by age three to four months, and chorioretinal lacunae (CRL). CRL is the most consistently observed trait in affected individuals [2,3,4]. In the absence of CRL, alternative eye phenotypes, such as coloboma and microphthalmia, satisfy the requirements for an AIC diagnosis provided that typical seizure types and malformation patterns are present [2]. This phenotypic heterogeneity emphasizes the need for identification of genetic causes to support the diagnosis of AIC [3] and guide prognostic and genetic counselling.

Early investigations into the genetics of AIC were predominantly focused on the X chromosome for three reasons [5,6,7,8,9,10]. Firstly, an X-linked, male-lethal cause best explains the greater than 99% female predominance of the disease; secondly, all affected males identified to date have a 47, XXY karyotype [11,12,13,14]; and thirdly, serendipitously the first chromosomal aberration reported in a girl with AIC was a 46,X,t(X;3)(p22;q12) translocation [15]. Despite 75 years of clinical and genetic investigations, a common X-linked cause is yet to be found. On reviewing the candidate variants reported in AIC thus far, there are no pathogenic variants reported in the same gene or sub-chromosomal overlaps (Table 1).

The clinical presentations of AIC in the reported cases with genetic investigations show considerable heterogeneity (Table 1). They comprise three groups: (i) the classical AIC triad (definite diagnosis), (ii) missing one or two classical features, with additional major features that satisfy the most recent proposed diagnostic criteria [2] (likely diagnosis) and (iii) missing two or more classical features with additional minor features that do not meet the most recent proposed criteria (suspected diagnosis). Based on the phenotypic heterogeneity of AIC, we hypothesized that individuals who have received an AIC diagnosis to date are likely to have genetically heterogeneous causes. Here, we aimed to perform unbiased, exome and/or genome sequencing analyses of individuals selected on the basis of their AIC diagnosis.

## 2. Materials and Methods

### 2.1. Human and Animal Ethics

All sequencing and recruitment to this study were carried out with approval from the Women’s and Children’s Hospital Human Research Ethics Committee (REC 2361/3/2014) and the Austin Human Research Ethics Committee. All zebrafish work in this study was performed under the University of Adelaide Office of Research Ethics, Compliance and Integrity Research Services, approval number S-2017-073. Mouse work in this study was performed under the University of Queensland Animal Ethics, approval number 2016/AE000486.

### 2.2. Exome and Genome Sequencing

All ES sequencing libraries (150 bp paired-ends) were captured using the Agilent SureSelect EZ human exome version 3 kit (Agilent Technologies, Mulgrave, Australia) and sequenced on the HiSeq2500 (Illumina, San Diego, CA) platform by the South Australian Cancer Council Genome Facility (Adelaide, Australia). GS was performed using the HiSeqXTen (Illumina) platform at the Kinghorn Centre for Clinical Genomics (Darlinghurst, Australia). Unaligned ES and GS reads were individually mapped to the hg38 (hs38DH) build of the human genome using BWA MEM [27]. All ES and GS alignments were processed for variant calling, annotation and analysis per Genome Analysis Toolkit (GATK) v4.1.9.0 best practice guidelines [28].

### 2.3. Candidate Variant Prioritization and Filtering

VCFs containing SNPs and indels identified from ES and GS were uploaded to Franklin (Genoox, https://franklin.genoox.com/ accessed on 15 May 2023) for annotation and detection of variants already classified as pathogenic or likely pathogenic in ClinVar. We additionally annotated variants using ANNOVAR [29] and then assessed and prioritized novel variants in alignment with the American College of Medical Genetics and Genomics and Association for Molecular Pathology (ACMG/AMP) guidelines [30].

Variants that lacked pathogenic evidence based on ACMG/AMP guidelines were prioritized for further investigation based on the following criteria: (i) residing in a known mutational hot spot or functional domain, (ii) located in evolutionarily conserved sequences, (iii) altered protein expression or localization, (iv) found in a variation-intolerant gene determined using RVIS [31] and/or previous publications, (v) in a gene implicated in eye or brain development with no current disease association, and (vi) supporting in silico evidence for pathogenicity by multiple algorithms, as previously defined [32].

### 2.4. Structural Variants

Structural variants (SV) larger than 50 bp were identified using DELLY [33], LUMPY [34] and Manta [35] for deletions, duplications, inversions, translocations and insertions and RetroSeq [36] specifically for mobile element insertions. Annotations for SV and mobile element insertions (MEI) were annotated using the BEDtools annotate function. Presumed benign SVs were filtered based on 500 bp regional overlaps with frequently observed SVs from the 1000 genomes project phase 3, healthy controls reported by Coe et al. [37], The Genomes of The Netherlands consortium [38], ISCA curated benign callset [39] and common variants detected from the Deciphering Developmental Disorders (DDD) study [40]. Putative disease-causing SV were selected as those containing at least one of the top 5% of predicted haploinsufficiency or triplosensitivity genes [41], known genes causing inherited retinal diseases (https://sph.uth.edu/RetNet accessed on 15 May 2023), disease-causing SV from the DDD study or ClinVar or genes implicated in Mendelian disease.

### 2.5. Molecular Modelling of Missense Variants

A predicted structure for WNT8B (based on UniProt accession number Q93098) was downloaded in protein databank (PDB) format from the SWISS-MODEL server https://swissmodel.expasy.org/repository/uniprot/Q93098 (accessed on 15 May 2023). The predicted structure of SLF1 (based on UniProt accession number Q9BQI6) was downloaded from the AlphaFold server https://alphafold.ebi.ac.uk/entry/Q9BQI6 (accessed on 15 May 2023) model AF-Q9BQI-F1. Prediction of missense variant effects were made using the DynaMut server with default options [42].

### 2.6. Plasmids and Cloning

Active WNT8B-V5 (WNT8B wild-type) was a gift from Xi He (Addgene plasmid # 43819; http://n2t.net/addgene:43819, (accessed on 15 May 2023; RRID:Addgene_43819) [43]. M50 Super 8× TOPFlash (Addgene plasmid # 12456; http://n2t.net/addgene:12456, (accessed on 15 May 2023; RRID:Addgene_12456) and M51 Super 8x FOPFlash (Addgene plasmid # 12457; http://n2t.net/addgene:12457, (accessed on 15 May 2023; RRID:Addgene_12457) were gifts from Randall Moon [44]. Renilla luciferase vector, pRL-TK plasmid, was obtained from Promega (Cat #E2241, Promega, Alexandria, Australia). The WNT8B p.L70P vector was cloned using a PCR insert, with a mutant sequence flanked by *Sac*I and *Apa*I sites ligated into the Active Wnt8B-V5 vector backbone. The PCR insert was generated by combining overlapping amplicons made with the following primer pairs: pair 1, WNT8B3F_43810 5′-AAGCAGAGCTCTCTGGCTAAC-3′ (*Sac*I underlined) and WNT8B2R_43742 5′-TGGCTGGACAGCTGC**G**GGGCTCTCTCAGGGC-3′ (mutant base bold, underlined); pair 2, WNT8B2F_43741 5′-GCCCTGAGAGAGCCC**C**GCAGCTGTCCAGCCA-3′ (mutant base bold, underlined) and WNT8B4R_43811 5′-CAGGGCCCGCCCGCGCCTTA-3′ (*Apa*I site underlined).

### 2.7. Cell Culture and Transfection

HEK293T cells were plated onto 12-well plates at a density of 0.5 × 10^6^ cells per well with 2 mL of growth media (DMEM, 10% FBS and 1% L-glut) and incubated at 37 °C in 5% CO_2_ air atmosphere for 15 h. Plasmids were transfected into HEK293T cells using Lipofectamine 2000 Transfection Reagent (Invitrogen, Cat # 11668027) according to the manufacturer’s recommendations except Lipofectamine 2000 Transfection Reagent was reduced to 2 μL per well, Opti-MEM medium was increased to 250 μL per well and cells were incubated with the transfection mix for 24 h at 37 °C in 5% CO_2_ air atmosphere. Transfections were performed with two technical replicates per condition in each experiment. Experimental groups were defined by co-transfection of the TOPflash or FOPflash (negative control) plasmid with the empty pcDNA3.1 vector, Active WNT8B wild-type, WNT8B p.L70P or a combination of WNT8B wild-type and WNT8B p.L70P. All transfections included 50 ng of the pRL-TK vector as a control for transfection efficiency.

### 2.8. TOPflash Dual Luciferase Assay

Reagents from the Dual Luciferase Reporter Assay Kit (Promega, Cat # E1960) were prepared following the manufacturer’s recommendation for 12-well plates. The luciferase reporter assay was performed according to the manufacturer’s protocol (Promega). The luminescence of each cell lysate was measured using a GloMax 20/20 Luminometer (Promega, Cat # E5311). The resulting firefly:renilla ratios were analysed by normalizing the relative luciferase activity (RLA) of each experimental group to the empty pcDNA3.1 vector control measurement.

### 2.9. Protein Extraction

Proteins were extracted from cultured cells in lysis buffer (120 mM NaCl, 0.5% Nonidet P-40 (NP-40), 50 mM Tris-HCl (pH 8.0), 1× protease inhibitor cocktail (Sigma-Aldrich, Cat # P8340, Macquarie Park, Australia), 1 mM NaF and 1 mM NaVO_4_) by incubation on ice for 15 min followed by homogenization via sonification (15 × 1 s pulses). Lysates were centrifuged at 15,000× *g* for 15 min at 4 °C, and the supernatants were retained. Protein concentration was determined using the “Pierce” bicinchoninic acid (BCA) assay (Thermo Fisher Scientific, Cat. # 23225, Scoresby, Australia) by following the manufacturer’s protocol.

### 2.10. Western Blot

Protein samples were denatured in 1× Laemmli buffer (0.0625 M Tris-HCl (pH 6.8), 2% sodium dodecyl sulphate, 10% glycerol, 5% β-mercaptoethanol and 0.001% bromophenol blue) at 95 °C for 2 min. Each sample (20 μg of total protein) was size fractionated on denaturing 7% polyacrylamide gels at 140 V for 1.5–2 h depending on the target protein size and transferred to BioTrace NT nitrocellulose membrane (Thermo Fisher Scientific Cat # 66485) using electroblotting. The membrane was blocked with 5% skim milk, 0.1% Tween-20 in Tris Buffered Saline (TBS) for 4 h at room temperature, and primary antibodies incubated with the membrane in blocking solution overnight at 4 °C. Membranes were washed 5 × 5 min in TBS, 0.1% Tween-20 at room temperature, and secondary antibodies were applied in blocking solution for 1 h followed by 5 × 5 min washes in TBS, 0.1% Tween-20. Antibody-bound membranes were treated with Amersham ECL Western blotting Detection Reagents (GE LifeSciences, Cat # RPN2109, Chicago, IL, USA) following the manufacturer’s protocol and exposed to Amersham Hyperfilm ECL at time intervals of 10 s, 30 s, 1 min and 5 min. Antibodies were used with the following concentrations: rabbit anti-β-Tubulin pAb (Abcam, Cat # ab21058, Cambridge, UK) at 100 ng ml^−1^, rabbit anti-V5 pAB (Bethyl Lab, Cat # A190-120P, Montgomery, TX, USA) at 500 ng ml^−1^, and secondary antibody, i.e., goat anti-rabbit IgG HRP conjugated (Agilent [Dako], Cat # P044801-2) at 330 ng ml^−1^.

### 2.11. Zebrafish Morphant Analyses

Adult Tübingen strain (Tu) zebrafish were housed in filtered and oxygenated fresh water, maintained at 28.5 °C and fed twice daily. Breeding was controlled by an automated 14 h light and 10 h dark cycle; embryos were collected via the marbling technique [45].

Morpholino antisense oligonucleotides (MO) were designed and synthesized by GeneTools (GeneTools, LLC., Philomath, OR, USA). All experimental MO were designed to block translation by targeting sequences upstream of the AUG start codon of the genes listed in Appendix A. A control MO that shares the same backbone as the experimental MO but has no target in zebrafish and minimal biological activity was included in each experiment as a negative control (Appendix A). Embryos at one- or two-cell stage were injected with MO [45] at a concentration gradient of 0.25 µM, 0.5 µM, 0.75 µM and 1 µM. The dose of MO that resulted in less than 50% of early embryonic death was selected for phenotype analysis. A total of 240 embryos were injected per experimental group for the final analysis.

### 2.12. Fetal Mouse Histology

In situ hybridization was performed as previously described [46]. Riboprobes for *Wnt8b*, *Slf1* and *Szt2* were generated using primers from the Allen Developing Mouse Brain Atlas (Website: © 2015 Allen Institute for Brain Science. Allen Developing Mouse Brain Atlas (Internet). Available from: http://developingmouse.brain-map.org accessed on 15 May 2023), (Appendix A). Each PCR amplicon was purified and cloned into pGEM^®^-T Vector System (Promega). The plasmids were linearised with *Not*I HF or *Sac*II restriction enzyme, (New England BioLabs, Notting Hill, Australia), purified (PCR Clean up Kit, QIAGEN, Clayton, Australia), transcribed using T7 or Sp6 RNA Polymerase, (New England BioLabs) and labelled with digoxigenin (DIG RNA labelling Mix, Roche, Millers Point, Australia) to generate the riboprobes. In situ hybridization was performed on coronal 50 μm vibratome sections or 20 μm frozen sections of wild-type CD1 mouse fetal brains.

Imaging was performed at the Queensland Brain Institute’s Advanced Microscopy Facility. Bright field imaging was performed with a Zeiss upright Axio-Imager Z1 microscope fitted with Axio-Cam HRc and HRm cameras, and images were acquired with Zen software (Carl Zeiss, Sydney, Australia). Images were cropped, sized and contrast-brightness enhanced for presentation with Photoshop and Illustrator (Adobe Systems, San Jose, CA, USA) and ImageJ (NIH, https://imagej.nih.gov/ij/download.html, accessed on 15 May 2023).

For detection of Wnt8b by indirect immunofluorescent staining, fetal brains were fixed in 4% paraformaldehyde (PFA) at 4 °C via transcardial perfusion. Either 20 μm frozen sections by cryostat or 50 μm vibratome were used. Sections were post-fixed with 4% PFA for 10 min and then antigen retrieved. The sections were then incubated in blocking solution (10% *v*/*v* normal donkey serum and 0.2% Triton X-100 in PBS) for 2 h at room temperature followed by incubation overnight with rabbit anti-WNT8B primary antibody at 1:500 (Thermo Fisher Scientific cat. # PA5-33117, RRID: AB_2550576). The next day, sections were washed with PBS and then incubated with a biotin-conjugated secondary antibody (Jackson Laboratories, Bar Harbor, ME, USA) used in conjunction with Alexa Fluor 647-conjugated Streptavidin (Thermo Fisher Scientific). The slides were then washed and stained for ten minutes with 0.1% 4′,6-Diamidine-2′-phenylindole dihydrochloride (DAPI) before being coverslipped with antifade mounting media.

Fluorescence images were captured using a Diskovery Spinning Disk confocal microscope (Nikon, Rhodes, Australia) equipped with a Zyla 4.2 sCMOS camera at the Queensland Brain Institute’s Advanced Microscopy Facility.

## 3. Results

### 3.1. Phenotypic Descriptions of Individuals Diagnosed with AIC

We studied 10 females from Australia (median age 8 years, range 18 months to 23 years), selected on the basis of an AIC diagnosis. One died at 18 months of age. Written informed consent was provided by the parents or legal guardians of all patients. The study was approved by the Austin Health Human Research Ethics Committee.

We reanalysed the phenotypes of each patient and classified their AIC diagnosis compared to current diagnostic criteria [2]. Five had classical AIC, two had likely AIC and three had suspected AIC (Table 1 and Appendix A). Five out of ten had no family history of seizures or developmental disorders; one had a brother with focal cortical dysplasia and intellectual disability, one had an aunt with seizures and one had a cousin with tonic–clonic seizures that were controlled with antiseizure medication (Table 2). The family history was not reported for two patients. No genomic or genetic abnormalities were detected on clinical testing including routine karyotype (1/10), chromosomal microarray (5/10) and candidate gene sequencing (1/10).

Seizures were present in all patients, with onset at a median age of 3.25 months (range 2 weeks–4 years). Initial seizure types were epileptic spasms in 7/10 and focal seizures in 3/10, and 6/10 patients had developmental delay prior to seizure onset. Epileptic spasms occurred in all individuals at some time. All showed developmental impairment after seizure onset, ranging from mild (one), moderate (two), severe (two) to profound (five). Six out of ten individuals regressed, and one had a diagnosis of autism spectrum disorder. Nine out of ten had CRL, and the one patient without had bilateral optic disc colobomas. ACC was present in 6/10; of the remaining four individuals without ACC, two had choroid plexus papillomas, one had thin corpus callosum and a posterior fossa arachnoid cyst and one had a thin corpus callosum with extensive cortical dysplasia. Scoliosis and vertebral abnormalities occurred in 6/10 patients. Two individuals required surgery for resection of cortical dysplasia and choroid plexus papilloma excision, respectively.

### 3.2. Exome and Genome Sequencing Excludes X-Linked Candidate Genes

We performed exome sequencing (ES) on 8/10 females with AIC. ES achieved a median coverage depth of 30× across 97.27% of the target capture (Appendix A). No individual had pathogenic or likely pathogenic variants from the ClinVar database (as of 14 November 2022) that could explain their phenotype. After prioritization (see Methods), three individuals had one unique variant each that was confirmed to be de novo by Sanger sequencing NM_032290.3 (*SLF1*): c.1409T > C: p.(Leu470Ser), NM_015284.3 (*SZT2*): c.9103C > T: p.(His3035Tyr) and NM_003393.3 (*WNT8B*): c.209 T > C: p.(Leu70Pro) (Table 3).

In the five trios without a plausible variant from ES and in two singleton individuals where ES was not performed, we proceeded with genome sequencing (GS). GS achieved a median coverage depth of 38× across mappable regions (99.74%) of hg19 (Appendix A). On average, approximately 5 million SNPs and indels were called per genome. For our first parse, we analysed variants in protein-coding regions that may have been missed by ES. On average, 42,000 exonic variants remained per proband and a further 82 fulfilled our disease-variant filtering criteria. For parent-proband trios (n = 5), we applied inheritance-based filtering, and only one de novo variant, NM_014727.1 (*KMT2B*): c.6418C > G: p.(Pro2140Ala), matched the filtering criteria and was validated via Sanger sequencing (Table 3). Given that more than 100 individuals with pathogenic KMT2B variants have been described in whom focal lower-limb dystonia is a distinguishing trait, this *KMT2B* variant was excluded from further analysis due to a lack of phenotypic overlap with other individuals with pathogenic variants in this gene [47,48]. For the two singletons, we compiled a list of filtered candidate variants with potential disease relevance, and none of these genes were recurrently affected in previous AIC sequencing cohorts or the present study. Individual 10 had a known pathogenic variant in *SMARCB1*, NM_003073.3(*SMARCB1*):c.1091_1093del (ClinVar: VCV000030201.8) implicated in Coffin–Siris Syndrome (CSS) [49] (Appendix A). Evaluation of the phenotype of Individual 10 considering this finding suggested CSS was the likely diagnosis, and the AIC diagnosis was removed. The presence of CRL in this individual may be a novel extension of the CSS phenotype.

We screened the GS for potential expansions of all known X chromosome short tandem repeats with repeat patterns 2–7 bp in length with ExpansionHunter [50] and surveyed all reads for potentially novel non-reference repeats with TRhist [51]; however, none of these analyses identified a repeat expansion. Finally, we looked for putative causative structural variants (SV) using DELLY [33], LUMPY [34], Manta [35] and RetroSeq [36]. The union of SV calls from the first three programs against truth sets for Genome-in-a-Bottle benchmark genome ‘NA12878′ [52] had a recall of 99% for deletions, 90% for duplications and 75% for break ends corresponding to other types of SV at the expense of precision of 51%, 48% and 42% for these events, respectively, necessitating additional filtering steps. We applied these programs to genomes from five proband-parent trios, two singletons and 100 unrelated individuals. Combining the variants from all SV callers, there were approximately 82,000 SVs per genome (Appendix A). SVs were filtered using the following parameters: not found in the 100 unrelated genomes, quality scores specific to each variant calling program and benign SVs from published datasets and assumed inheritance models (Appendix A). We further limited our analysis to variants that flanked hg19 canonical transcripts or regulatory elements, and none of these variants appeared de novo in silico (Appendix A).

### 3.3. The WNT8B p.Leu70Pro Variant Has a Dominant Negative Effect on Wnt Signalling

To assess the effect of the *WNT8B* (NM_003393.3:c.209T > C:p.Leu70Pro) variant found in patient 1 (Table 1; Figure 1a) on canonical WNT signalling, we performed a TOPflash assay in human embryonic kidney 293T cells [44]. In the presence of wild-type WNT8B, the mutant had a dominant negative effect indicated by a significant reduction in luciferase reporter gene activity (Figure 1a). We observed that the WNT8B mutant was consistently lower in abundance compared to the wild-type at equivalent levels of transfected plasmid, potentially indicating a loss of protein stability (Figure 1b). Modelling the effect of introduction of the p.Leu70Pro variant using DynaMut [42] showed a change in free energy between the predicted mutant to the wild-type structures of (ΔΔG) −1.25 kJmol^−1^, suggesting a loss of structural stability (Figure 1c,d). *WNT8B* has yet to be implicated in any human inherited disorder; however, the demonstrated roles for zebrafish and mouse *WNT8B* orthologues in specifying the anterior neuroectoderm, regulating axon guidance and specification of retinal progenitor cells suggest that this gene is a highly plausible candidate to explain disease in patient 1 and therefore a possible gene for AIC [53,54,55,56].

### 3.4. Knockdown of wnt8b and slf1 in Zebrafish Leads to Altered Eye and Body Development

We used morpholino knockdown in zebrafish (*Danio rerio*) embryos to screen for phenotypes that were suggestive of AIC for *SLF1* and *SZT2* genes because they did not have established molecular or biochemical assays and are yet to be implicated in both eye and brain development. We included wnt8b and tead1 (the only gene previously implicated in AIC [23] that is also conserved in zebrafish) for comparison. The morpholino injections and data analysis were carried out by researchers that were blinded to the conditions, and morphant classification counts were verified by three independent researchers. More than 90% of the uninjected and control morpholino groups displayed normal embryonic development with or without minor body abnormalities such as a bent tail (Figure 2a–c). In contrast, morphants for *tead1*, *wnt8b*, *szt2* and *slf1* all displayed a spectrum of developmental abnormalities including lack of eye pigmentation, body curvature, pericardial oedema, tail defects and head malformations (Figure 2a–c). The most consistent phenotype among the *slf1*, *tead1* and *wnt8b* morphants was the lack of eye pigmentation, which was patchy in appearance and often unilateral (Figure 2a). A lack of eye pigmentation was also observed in some *szt2* morphants; however, as 72 h post fertilization (hpf) *szt2* morphants often resembled 48 hpf morphants, this was attributed to developmental delay. There are known limitations to disease modelling in zebrafish [57] and the morpholino knockdown approach models a loss of function whereas our de novo variants may be dominant negative. Nonetheless, we showed a unifying morphant phenotype of AIC-like eye and brain defects among at least three of the four genes tested that was distinct from control morpholino phenotypes (Figure 2c).

Given the restricted nature of disease target tissues in AIC, we sought further evidence to implicate our AIC candidate genes based on their expression pattern in the prenatal eye and brain. We examined the developmental expression pattern of *Wnt8b*, *Slf1* and *Szt2* in the mouse brain at embryonic days E12 (Figure 3) and E14 (Appendix A). At E12, Wnt8b was expressed along the medial telencephalon, notably in the cortical hem, roof plate and the thalamic eminence. In the developing diencephalon, Wnt8b was expressed along the ventricular zone of the hypothalamus (Figure 3a). *Slf1* was expressed with an increasing gradient from anterior to posterior in the neocortex and the ventricular zones of the medial and lateral ganglionic eminences at E12. In the diencephalon, *Slf1* expression was restricted to the thalamic nucleus (Figure 3b). By E14, Wnt8b expression was diminished compared to earlier stages, as noted previously [59], while *Slf1* showed robust and restricted expression in the neocortex, piriform cortex, claustrum, amygdala and hippocampus (Appendix A). In contrast to Wnt8b and *Slf1*, *Szt2* was expressed diffusely throughout the developing brain at both E12 and E14 (Figure 3c and Appendix A).

## 4. Discussion

In 10 individuals with classical (5), likely (2) or suspected (3) AIC, we identified rare and predicted damaging variants in four autosomal genes (*WNT8B*, *SLF1, KMT2B* and *SZT2*) in four unrelated individuals. There was also a known pathogenic variant in *SMARCB1* in a fifth atypical case, where AIC was only suspected, that provided an alternate diagnosis of Coffin–Siris syndrome. The candidate genes identified by others and our genetic studies to date suggest that, in at least some cases, AIC does not fit an X-linked male-lethal inheritance model. We acknowledge, however, that the majority of individuals remain genetically undiagnosed and the extraordinary female bias in AIC strongly suggests an X-linked cause that may not have been detected in genetic studies to date due to technical limitations or a novel X-linked mechanism. In this study, short-read, paired-end ES and GS may have missed variants in highly repetitive regions of the genome and larger or complex structural variants. Our 38X average sequencing coverage of peripheral blood DNA could not detect somatic mosaic variants with an allelic fraction less than 5% nor brain-specific mosaicism. The significance of non-coding variants in regions lacking functional annotations could potentially be resolved with RNA-Seq of patient-derived cell lines or tissues or detection of epigenetic signatures. For the remaining unresolved individuals in this cohort, an X-linked cause due to mosaic or non-coding variants is possible.

The most enigmatic aspect of AIC is why it is almost exclusively a female disease. Considering our genetic data and that from prior investigations (Table 1) that opposes the X-linked male-lethal hypothesis, we explored other mechanisms that can lead to female-biased traits. One possibility would be the participation of the implicated AIC genes in a sexually dimorphic pathway such as Wnt signalling, which has been linked to female developmental disorders in X-linked, Wnt-related genes such as *DDX3X*, *USP9X* and *PORCN* [60,61,62]. In mice, the Wnt signalling pathway directs sexually dimorphic autosomal gene expression in specification of ovaries and testis [63] and in the adrenal cortex, activation of Wnt signalling by Rspo1 results in adrenal cortical hyperplasia in females but cortical thinning in males [64]. We identified a *WNT8B* variant that affected Wnt signalling as measured using the TOPflash assay; while Shrauwen et al. showed upregulation of the Wnt receptor LRP5 using RNA-Seq, and also discovered a pathogenic variant in *TEAD1* [23]. While TEAD1 is involved in the hippo pathway, there is evidence for cross-regulation between the hippo and Wnt/β-catenin pathways [65]. Another hypothesis for the female preponderance would be the interaction with, or modulation by, female sex hormones and candidate AIC genes. The role of estrogen in retinal pathology is well established and it is tempting to speculate that CRL, the pathognomonic feature of AIC, may be a sex-limited trait that appears in more females than males therefore biasing the diagnosis [66,67]. Transcriptomic and epigenomic data will be needed to assess whether estrogen directly or indirectly regulates implicated AIC genes.

Based on our in vivo findings and murine expression studies, the de novo *WNT8B* and *SLF1* variants that we discovered appear to be good candidates to explain AIC in these individuals. These two genes have yet to be implicated in human neurodevelopmental disorders, and we could not identify additional individuals with phenotype and genotype overlap using GeneMatcher [68] or other AIC cohorts sequenced to date [23,24,69]. Knockdown of *tead1*, *wnt8b* and *slf1* produced similar zebrafish morphant phenotypes, with the highest frequency of eye-related defects observed in *wnt8b* morphants (Figure 2c). The defects in our *tead1*, *wnt8b* and *slf1* morphants demonstrate the involvement of these genes in the development of tissues (eye and brain) in which AIC pathologies are observed. *Wnt8b* and *Slf1* had similar expression patterns in embryonic mouse brain (Figure 3 and Appendix A). Both individuals with the *WNT8B* and *SLF1* variants displayed the classical triad of AIC with accompanying features including ventriculomegaly, atrophy of the cerebellar hemispheres, scoliosis and skeletal abnormalities (Table 2 and Appendix A). *SZT2* and *KMT2B* are less convincing candidates for AIC. Both genes are implicated in other syndromes and their expression is not restricted to critical target tissues in the embryonic brain as it is for *WNT8B* and *SLF1*. Taken together, our findings and that of others [23,24] suggest that the causative alleles in individuals diagnosed with AIC will not be all X-linked. When one considers that the AIC spectrum is wide and includes classical, likely and suspected cases, each with variable clinical and imaging phenotypes within each subgroup, it is likely that the genetic heterogeneity we identified is reflective of AIC not being a single malformation syndrome. Ongoing work to improve the phenotypic classification of AIC [70,71] are likely to assist with future attempts at gene identification.

## Figures and Tables

**Figure 1 genes-14-01565-f001:**
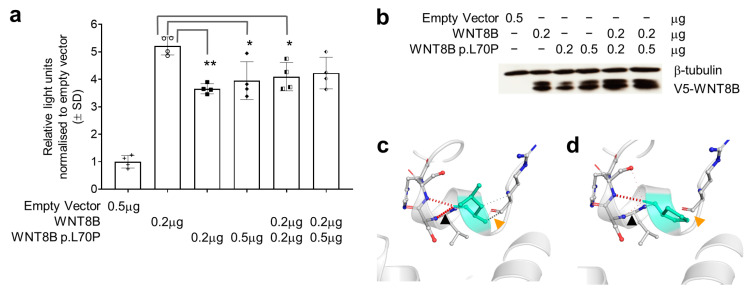
WNT8B p.Leu70Pro (p.L70P in this figure) affects protein function and structure. (**a**) TOPflash dual-luciferase reporter gene activity in HEK293T cells transfected with pcDNA3.1+ (empty vector) or pcDNA3.1+ expressing cDNAs for wildtype WNT8B, the p.Leu70Pro variant or combinations of both as shown below the horizontal axis. Data in the bars represent the mean and standard deviation of the ratio of firefly to *Renilla* luciferase luminescence normalised to the mean luminescence ratio in cells transfected with the empty vector from two independent transfections, each with technical duplicates per group. Individual replicate data are shown with point symbols for each bar. Comparisons were made using one-way ANOVA between all groups except the empty vector with significant differences when assessed using Tukey’s post hoc tests observed as shown * *p* < 0.05, ** *p* < 0.01. (**b**) Western blot showing the abundance of WNT8B wildtype and p.Leu70Pro variant proteins detected using rabbit anti-V5 antibody 500 ng/mL (Bethyl Cat# A190-120P, Montgomery, TX) in each transfection relative to β-tubulin detected using rabbit anti-β-tubulin antibody directly conjugated to HRP 200 ng/mL (Abcam Cat# ab21058, Cambridge, United Kingdom). (**c**) WNT8B compared to the predicted changes (**d**) in structure due to the p.Leu70Pro variant as assessed using DynaMut. Note that within the vicinity of the p.Leu70Pro variant there is loss of hydrogen bonding (black arrows) and loss of non-polar interactions (orange arrows).

**Figure 2 genes-14-01565-f002:**
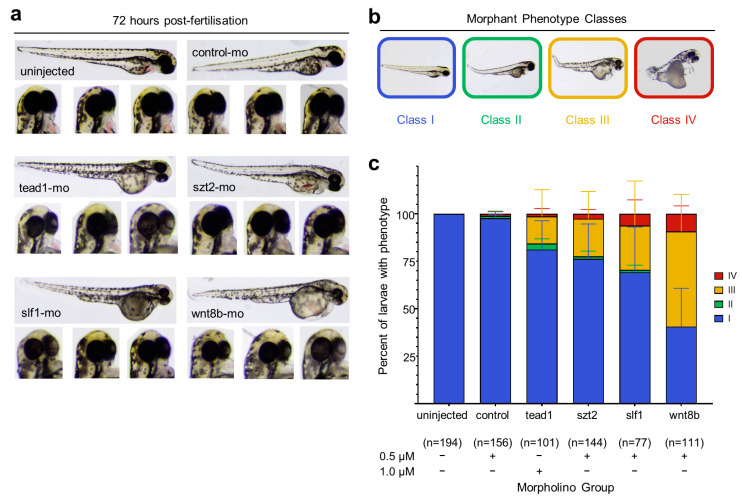
Phenotype analysis of morpholino mediated knockdown of Aicardi candidate genes in zebrafish embryos. (**a**) Examples of morphology of zebrafish embryos at 72 h post fertilization (hpf) either uninjected or injected with a control- or gene-targeted morpholino. Note the reduction in eye pigment in embryos injected with the tead1, slf1 or wnt8b morpholinos. (**b**) Zebrafish embryos were scored according to the classification system of Miesfeld et al. (2015) [58]. Class I: normal phenotype; Class II: normal eye pigmentation with mild body defects; Class III: reduced eye pigmentation with normal to mild body defects; Class IV; reduced eye pigmentation and severe body defects. (**c**) Observed morphant phenotypes after researcher-blinded injections of 240 zebrafish embryos per group. The number of viable embryos remaining after 72 hpf are denoted by (n). The amount of morpholino injected per group are indicated by (+).

**Figure 3 genes-14-01565-f003:**
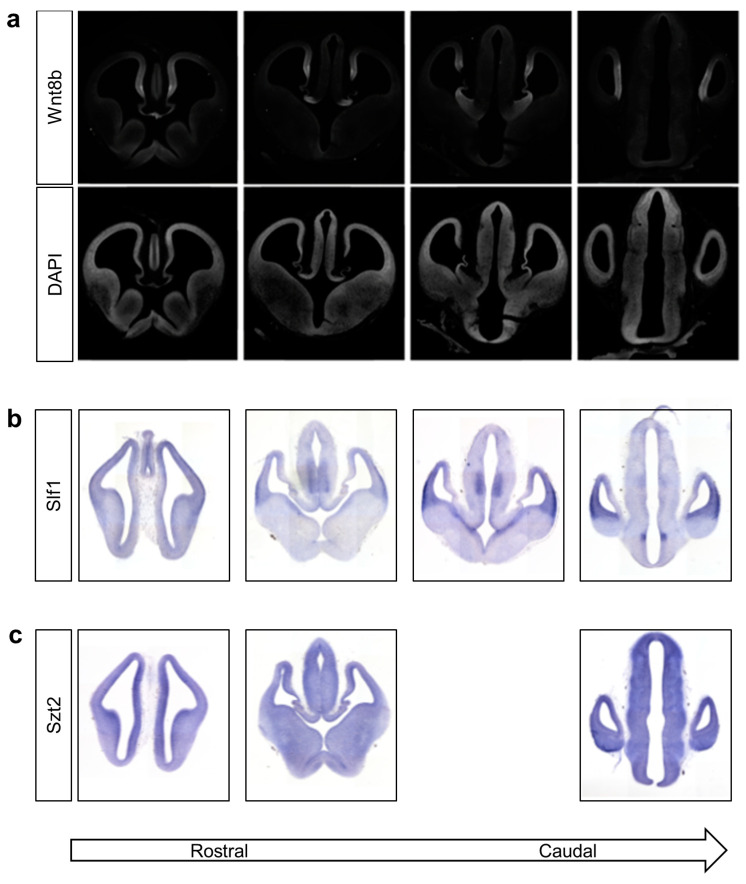
Wnt8b and Slf1 have spatially restricted expression patterns in the developing mouse brain. Coronal 20 μm frozen sections of mouse brain harvested at embryonic day 12. (**a**) Top row shows expression of Wnt8b measured using indirect immunofluorescent staining. Detection by rabbit anti-WNT8B primary antibody. Nuclei in the same section visualised using DAPI staining are shown in the second row to define morphology of the section. (**b**) Detection of *Slf1* transcripts by in situ hybridisation using an in vitro transcribed anti-sense riboprobe. (**c**) Detection of *Szt2* transcripts by in situ hybridisation using an in vitro transcribed anti-sense riboprobe.

**Table 1 genes-14-01565-t001:** Variants reported in published individuals with Aicardi Syndrome.

Genetic Variant [Reference]	Typical AIC Traits	Atypical AIC Traits	Diagnosis
46,X,t(X;3)(p22;q12) [15]	Scoliosis, Rib anomalies, Reduced muscle tone, Bilateral ptosis, Microopthalmia, Severe DD	Symblepharon, Corneal Ulceration, Lagopthalmus, “Clods” of retinal pigment	Suspected
46,XX,t(12;21)(q13.3;q11.2) [16]	CRL, Coloboma, ACC, Infantile spasm, Hypsarrhythmia		Definite
Xp22.2pter Deletion and Partial 3p Trisomy [17]	Micropthalmia, Ventriculomegaly	Sclerocornea, Posterior hair whorl, Teratoma/lipoma in optic chiasm, Sharply demarcated chorioretinal defects, No ACC, Costovertebral anomalies, Infantile spasms or Intracranial heterotopias	Suspected
46,X,t(X;7)(p22.3;p15) disrupting *CDKL5* [18]	CC hypoplasia, Infantile spasm	Pale fundi, Microcephaly, Choreoform and Myoclonic dyskinesia	Suspected
1p36pter Deletion [19]	Infantile spasm, bilateral papillary coloboma, ACC Ventricular dilatation, Delayed psychomotor milestone	Brachydactyly, Hypertrichosis, Deep set eyes, Posterior rotated ears	Phenocopy
6q27qter Deletion (includes *DLL1*) and 12q24.32q24.33 Duplication [20]	Neurodevelopmental delay, Infantile spasm, Partial ACC, Colpocephaly, Ventriculomegaly, Gross cerebral asymmetry, Other cortical malformations, Coloboma, CRL, Scoliosis, Craniofacial abnormalities		Definite
3q21.3q22.1 Deletion [21]	Preaxial polydactyly, Muscular hypotonia, Abnormal corpus callosum, Chorioretinal lacunae, Partial ACC, Ventricular dilatation, Mild cortical thickening, Cavum septum pellucidum	Retromicrognathia, Atrial-septal defect, Mild pulmonary valvular stenosis	Suspected
45,X0/46,XX [22]	Ventricular dilatation, Intraventricular cysts, Muscular hypotonia, Optic nerve coloboma, ACC, Intracranial cysts, Multiform epileptic seizures, Chorioretinal atrophy	Short neck, Pterigium colli, Teletelia, Barrel-shaped thorax, Lymphedema, Hemispondylia, Patent foramen ovale, Minimal persistent arterial duct, Hydrocephalus, Intracranial hypertension, Deafness	Aicardi and Turner Mosaic
NM_021961.5 (*TEAD1*):c.618G > A; p.Trp206Ter [23]	CRL, Infantile spasm, Cerebellar cysts, Periventricular heterotopias		Definite
NM_024578.1 (*OCEL1*):c.499G > A; p.Ala167Thr [23]	Partial ACC, CRL, Infantile spasm, Posterior fossa arachnoid cyst		Definite
NM_002518.4 (*NPAS2*):c.2465C > T; p.Pro822Leu [24]	CRL, Infantile spasm, ACC		Definite
NM_032656.3 (*DHX37*):c.1145A > G; p.Asp382Gly [25]	CRL, Infantile spasm, ACC, Coloboma, Colpocephaly, Ventriculomegaly, Intracranial cysts, Polymicrogyria, Heterotopia, Severe DD and ID, Dysmorphic features		Definite
Xp22.33 duplication (*SHOX*) [26]	CRL, Infantile spasm, Thin corpus callosum, Frontal polymicrogyria, Ventricular dysmorphism, Seizures		Definite

Abbreviations: ACC, agenesis of the corpus callosum; CRL, chorioretinal lacunae; DD, developmental delay; ID, intellectual disability.

**Table 2 genes-14-01565-t002:** Clinical summary of the cohort.

Proband ID	1	2	3	4	5	6	7	8	9	10
Candidate variant	NM_003393.3 (*WNT8B*):c.209T > C:p.(Leu70Pro)		NM_032290.3 (*SLF1*): c.1409T > C: p.(Leu470Ser)					NM_015284.3 (*SZT2*): c.9103C > T: p.(His3035Tyr)	NM_014727.1 (*KMT2B*): c.6418C > G: p.(Pro2140Ala)	NM_003073.3(*SMARCB1*):c.1091_1093del
Aicardi diagnosis	Classical	Classical	Classical	Classical	Classical	Likely	Likely	Suspected	Suspected	Suspected
Evidence for AIC diagnosis	ACC, CRL, Spasms	ACC, CRL, Spasms	pACC, CRL, Spasms	ACC, CRL, Spasms	ACC, CRL, Spasms	CRL, Spasms, Cortical Malformation (Mj), Intracranial cyst (Mj). Two classical, two major.	Spasms. Cortical Malformation (Mj) CPP (Mj), Optic disc coloboma (Mj), microphthalmia (min). One classical, three major, one minor.	CRL, Spasms. Cortical Malformation (Mj). Two classical, one major.	CRL, Spasms, CPP (Mj), Vertebral and costal abnormalities (Min). Two classical, one major, one minor	ACC, CRL, Scoliosis (min). Two classical, one minor.
Sex	F	F	F	F	F	F	F	F	F	F
Age at latest examination	9 years	17 years	23 years	11.5 years	4 years	8 years	19 months (deceased)	7 years	23 years	7.5 years
MRI	ACC and interhemispheric cyst; PNH; subcortical heterotopia; PMG	ACC; choroid plexus cysts; PNH	pACC	ACC	ACC; PMG; subcortical heterotopias; septated cyst beneath L cerebellar hemisphere	Thin CC; PMG; posterior fossa arachnoid cyst	Thin CC; cortical dysplasia; bilateral CPP	Thin CC; extensive cortical dysplasia	Bilateral CPP	Corpus callosum dysgenesis,
Ocular findings	Bilateral CRL, strabismus, nystagmus, astigmatism, hypermetropia	Bilateral CRL, left optic nerve coloboma	Bilateral CRL, exotropia, hypermetropia	CRL, peri-papillary chorioretinal atrophy	Bilateral CRL, left optic disc coloboma	CRL	Bilateral choroid-retinal abnormalities, bilateral optic disc coloboma, focal nodular retinal thickening on the left, bilateral microphthalmia, nystagmus and downward gaze	Unilateral CRL (left eye only)	Bilateral CRL, pigmented dystrophic right disc and mild tilt of left disc	Unilateral CRL (right eye only), high myopia
Intellectual disability	Profound (non-verbal)	Profound (non-verbal)	Profound: At 12 years; 4 words. Single words; 2 years	Profound (non-verbal)	Mild/moderate: 3 months fix and follow, 5 months—grabbing, babbling, 15 months no words, 17 months “dad” but not specific.	Severe: At 8 years; non-verbal	Profound.	Moderate; At 21 months, understood 5 words, said “ba” “ma”. 27 months, babbling but no words. 7 years: talking in sentences of 6–8 words, can read and write.	Mild	Severe 3 years: non-verbal, 7.5 years: Special school, one word only.
Motor	Non-ambulatory, spastic quadriplegia	Non-ambulatory, hypotonia	Walks with supervision. Sat 17 months, walked 4 years	Hypotonia. Minimal independent mobilisation, waddling gait, decreased tone and reflexes in upper limbs, paucity of movements	At 5 months rolling, 6 months delayed head control, rolling front to back but not back to front, 8 months pulling up, 17 months taking steps with walker.	At 2 years 8 months, not crawling, able to sit. At 8 years, able to walk assisted	At 10 months trying but unable to roll, can only lift head for a brief period	At 21 months, still learning to pass objects between hands, can pick up small objects but not hold a pencil correctly, she could throw a ball and take steps forward but not backwards nor kick a ball. Able to run at 7 years	Sat at 9 months. Walked at 15 months	At 3 years, sitting, not walking, rides a tricycle. At 7.5 years: commando crawling, dependent for all activities of daily living, not toilet trained.
Development prior to seizure onset	Abnormal	Abnormal	Abnormal	Abnormal	Normal	Normal	Delayed	Normal	Normal	N/A
Development after seizure onset	Delayed	Delayed	Delayed	Delayed	Delayed	Delayed;	Delayed	Delayed	Mild delay	N/A
Autism Spectrum Disorder	No	No	No	No	No	No	No	Yes	No	No
Developmental Regression	Yes. Regressed with introduction of clonazepam at 5 months	Yes. Stopped rolling 18 months, lost single word at 2 years, further loss of motor milestones and interaction	Yes. Gait deteriorated at 19 years	Yes. Loss of smiling at seizure onset (2–3 months)	No	Yes at seizure onset	No	Yes. Anecdotally began to progress when she came off medication.	No	No
Seizure onset age	2 weeks	3.5 months	4 years	2 months	5 months	10 weeks	5 months	3 months	6 months	N/A
Initial seizure type	Right facial clonic	Infantile spasms	Focal Status	Infantile spasms	Infantile spasms	Hemiclonic	Infantile spasms	Asymmetric infantile spasms	Predominantly right sided Flexion Spasms	N/A
Seizure types	Infantile spasms, FBTC, tonic seizure, focal motor seizures, >3 per week.	Infantile spasms, TCS, tonic seizures, myoclonic seizure, absence, atonic, hemiclonic	Infantile spasms (ongoing at 23 years), focal seizures, febrile seizure at 1 year	Infantile spasms, FBTC, TCS, focal seizures, myoclonic seizures, gelastic, SE, atonic	Infantile spasms, focal seizures	Infantile spasm, SE, hemiclonic, head drops	Infantile spasms, focal tonic.	Infantile spasms, tonic seizures	Infantile spasm, TCS, possible aura before the infantile spasms, SE, PNES	N/A
EEG	Multifocal discharges. Electroclinical focal seizure captured at left central region.	Multifocal discharges. Tonic and atonic seizures recorded	Bilateral fronto-temporal and fronto-central independent sharp and sharp-slow waves, predominantly from the left. Spasms captured.	Multifocal discharges, modified hypsarrhythmia. Tonic spasms recorded.	Epileptiform activity over left posterior-temporal region, modified hypsarrhythmia. Spasms recorded.	Predominant epileptiform activity from left frontal region, with generalised spread, hypsarrhythmia	Posterior spike/slow discharges, biposterior sharp activity and bursts of posterior fast activity. High amplitude delta slowing over bilateral posterior quadrants and infrequent sharp slow waves over the left occipital regions. Focal tonic seizure captured.	Left frontal and anterior temporal activity. Tonic, focal and spasms captured on EEG, modified Hypsarrhythmia	Focal discharges, most prominent in right posterior quadrant and left frontal regions. Bitemporal discharges. Generalised bursts of polyspike wave and irregular spike and slow wave. Spasms captured with associated with slow wave discharges. Hypsarrhythmia	Normal
Family history of seizures	No	Yes (maternal second cousin with well controlled TCS)	No, brother (who does not carry the *SLF1* variant) with FCD and ID	No	No	No	No	No	No	No
Other	PEG, scoliosis, dental caries, drooling, hip dysplasia, pelvic deformity, microcephaly	PEG, scoliosis, failure to thrive, irritability, constipation, drooling, chronic pain, hip dislocation	Scoliosis, Becker-Nevus Syndrome, Poland Syndrome, clinodactyly, syndactyly, short forearm, hypopigmentation, skin tags, hirsute patches, facial dysmorphism, short stature, sleep apnea, torticollis, vertebral fusion	Asthma, dental caries, constipation, drooling, finger sucking, excoriation of her fingers and palms, facial features: short philtrum, upturned nasal tip and somewhat large pinnae.	Vertebrate abnormalities, torticollis, physio from 2 months.	Hemiparesis, hypotonia, nephrocalcinosis, hypermobile joints	Subdural shunt, naso-gastric tube	Hypotonia, mild facial dysmorphisms	Scoliosis, depression	Bilateral hip dysplasia, scoliosis, pectus excavatum, growth retardation, previous PEG, conductive hearing issues, delayed dentition, sparse scalp hair, hirsutism, thin upper lips, small ears, absent/hypoplastic nail 5th toes, 4th toe clinodactyly, 2nd toes override 3rd toes, 2× café au lait macules

ACC, agenesis of the corpus callosum; CRL chorioretinal lacunae; Min, minor; Maj, major; CPP, choroid plexus papilloma; PNH, periventricular nodular heterotopia; PMG, polymicrogyria; ID, intellectual disability; FCD, focal cortical dysplasia; FBTC, focal bilateral tonic–clonic seizure; TCS, tonic–clonic seizure; EEG electroencephalogram; PNES, psychogenic non-epileptic seizures; SE, status epilepticus; PEG, percutaneous endoscopic gastrostomy.

**Table 3 genes-14-01565-t003:** Summary of novel candidate variants.

Individual	Symbol	hg38	RefSeq	cDNA Variant	Protein Variant	PolyPhen2	CADD	GERP+	gnomAD v2.1.1	DynaMut ΔΔG (kJmol^−1^)	Zygosity	Inheritance
8	*SZT2*	chr1:43447156	NM_015284.3	c.9103C > T	p.His3035Tyr	0.997 (D)	18.18	5.43	0	No available structure	Heterozygous	De novo
3	*SLF1*	chr5:94665901	NM_032290.3	c.1409T > C	p.Leu470Ser	0.946 (D)	21.10	5.41	0	−13.832	Heterozygous	De novo
1	*WNT8B*	chr10:100479980	NM_003393.3	c.209 T > C	p.Leu70Pro	0.962 (D)	21.20	5.70	0	−1.251	Heterozygous	De novo
9	*KMT2B*	chr19:35732967	NM_014727.1	c.6418C > G	p.Pro2140Ala	0.677(P)	15.08	3.99	0	No available structure	Heterozygous	De novo

## Data Availability

Exome and genome sequencing data are available from the corresponding authors for projects approved by the Women’s and Children’s Health Network human research ethics committee. Data corresponding to luciferase assays presented in Figure 1 and zebrafish morphology presented in Figure 2 are provided with this manuscript.

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
