# Peer review of "Aicardi Syndrome Is a Genetically Heterogeneous Disorder"

_genes, 2023, doi:10.3390/genes14081565_

Round 1

Reviewer 1 Report

 1. In the previous descriptions of Aicardi syndrome, several patients had important associated features including benign and malignant neoplasia (hepatoblastoma, angiosarcoma, teratoma, embryonal carcinoma) and precocious puberty. Have any of the individuals included in the study presented any similar disturbances?

2. I suggest authors to include the Supplemental Table A1 content in the manuscript as it is important for the Discussion of their results.

3. Several of the genes tested and which disclosed suspected variants have not been previously associated with corresponding phenotypes, including KMT2B (PM2 and PP2 criteria; VUS), SLF1 (PM2 criteria; VUS), SZT2 (PM2 criteria; VUSand WNT8B (10q24.31; PM2 and PP3 criteria; VUS). Among these genes, mainly WNT8B has a possible correlation directly related to a complex neurodevelopmental phenotype, due to is expression during early forebrain development and during neural regionalization. 

Author Response

We thank the reviewer for their comments, please find our responses below.

  1. In the previous descriptions of Aicardi syndrome, several patients had important associated features including benign and malignant neoplasia (hepatoblastoma, angiosarcoma, teratoma, embryonal carcinoma) and precocious puberty. Have any of the individuals included in the study presented any similar disturbances?

Phenotypes are as reported in Table 1. Two had choroid plexus papillomas (individuals 7 and 9) no other neoplasms were identified. None had precocious puberty.

  1. I suggest authors to include the Supplemental Table A1 content in the manuscript as it is important for the Discussion of their results.

We have moved Supplemental Table A1 to the main manuscript as Table 1.  All Supplemental tables and manuscript tables have been renumbered.

  1. Several of the genes tested and which disclosed suspected variants have not been previously associated with corresponding phenotypes, including KMT2B (PM2 and PP2 criteria; VUS), SLF1 (PM2 criteria; VUS), SZT2 (PM2 criteria; VUS) and WNT8B (10q24.31; PM2 and PP3 criteria; VUS). Among these genes, mainly WNT8B has a possible correlation directly related to a complex neurodevelopmental phenotype, due to is expression during early forebrain development and during neural regionalization.

We agree with the assessment of the reviewer that given lack of replication in additional affected individuals that these variants must all be considered as VUS. We agree that WNT8B is a very plausible candidate gene for AIC given the expression pattern and role in early forebrain development.

Reviewer 2 Report

Aicardi syndrome (AIC) is a rare genetic disorder that is defined by three cardinal features: agenesis of the corpus callosum, chorioretinal lacunae, and infantile spasms. This disorder singularly affects females giving rise to a hypothesis that the disease may be X-linked. However, the genes responsible for AIC are yet unknown. This study explored the genetic differences among 10 females with suspected or confirmed AIC through exome and genome sequencing. The de novo variants obtained from the sequencing analysis were tested for developmental expression in fetal mice brains and the effects of these mutations on neurodevelopment were studied using morpholino knockdown in zebrafish embryos. The study showed de novo mutations in KMT2B, SLF1, SMARCB1, SZT2, and WNT8B in five female patients, of which, Wnt8b and Slf1 showed relevant cortical expression in the mouse embryo brain and led to a greater number of zebrafish embryos with poor brain and ocular development.

The study suggests that multiple genetic mutations may be responsible for AIC and that some genes involved in the disease may not be X-linked.

Overall, AIC is a rare disorder and there is a lack of information about the etiology of the disease, this study provides an important piece of evidence about the genes that may be involved in the development of AIC/

Specific comments

  1. The authors have not clearly discussed Figure 2c in the results or in the discussion section.
  2. Figure 3: In the figure legend the authors talk about temporal expression of Wnt8 and Slf1 however, they only show data at one time point – E12, please clarify.

a)     Figure 3c: The third section towards the caudal region is not visible in the image, please comment if there is no Szt2 staining in the section.

b)     The authors performed IHC for Wnt8 suggesting protein level and ISH for Slf1 and Szt2 suggesting transcript levels. Since transcript expression level does not always correlate with the expression of the protein, the use of one technique to study the expression pattern across all three proteins/transcripts would be ideal to interpret.

  1. Line 435-438: In these lines, the authors present a rationale as to why AIC affects females even though the genes that may be involved in the disease are not X-linked. The authors should consider expanding more on this rationale as it is difficult to understand the statement in light of X-linked gene mutations involving DDX3X, USP9X, and PORCN mentioned in the discussion.
  2. The nomenclature of properly denoting genes and proteins should be corrected in the manuscript for example in Fig 3, SLF1 should be italicized when mentioning the transcript level.

Author Response

We thank the reviewer for their comments, please find our responses below.

  1. The authors have not clearly discussed Figure 2c in the results or in the discussion section.

We have corrected the typographical errors in referencing this figure in the results and added to the discussion.

  1. Figure 3: In the figure legend the authors talk about temporal expression of Wnt8 and Slf1 however, they only show data at one time point – E12, please clarify.

The reviewer is correct, we only showed one time point in this figure and we have removed the reference to temporal expression.  A second time point at E14.5 appears in Supplemental Figure A2.

  1. a) Figure 3c: The third section towards the caudal region is not visible in the image, please comment if there is no Szt2 staining in the section.

We do not have a matching section for Szt2 in this series to compare with other genes we tested.  We have revised this figure to make it clearer that the matched panel is not available for Szt2.

  1. b) The authors performed IHC for Wnt8 suggesting protein level and ISH for Slf1 and Szt2 suggesting transcript levels. Since transcript expression level does not always correlate with the expression of the protein, the use of one technique to study the expression pattern across all three proteins/transcripts would be ideal to interpret.

We could not obtain reliable antibodies for Slf1 and Szt2 therefore we used ISH for these two genes.

  1. Line 435-438: In these lines, the authors present a rationale as to why AIC affects females even though the genes that may be involved in the disease are not X-linked. The authors should consider expanding more on this rationale as it is difficult to understand the statement in light of X-linked gene mutations involving DDX3X, USP9X, and PORCN mentioned in the discussion.

We thank the reviewer for this suggestion and we add broader examples of the roles of WNT/β-catenin signalling in sexually dimorphic control of expression of autosomal genes.  While these examples are not brain specific, they serve to show that the Wnt/β-catenin pathway can be differentially regulated in males and females.

  1. The nomenclature of properly denoting genes and proteins should be corrected in the manuscript for example in Fig 3, SLF1 should be italicized when mentioning the transcript level.

Nomenclature has been corrected throughout.

Reviewer 3 Report

Authors applied exome and genome sequencing of 10 females with diagnosed, likely and suspected AIC, identified five genes KMT2B, SLF1, SMARCB1, SZT2 and WNT8 in five individuals, indicating that AIC is a genetically heterogeneous disorder.

1. Could authors discuss more about: why AIC has a strong heterogeneous feature but only 4000 individuals are affected with AIC? 

2. AIC is diagnosed as a multiple mixed phenotypes, including agenesis of the corpus callosum, epileptic spasms and so on, could KMT2B and SMARCB1 also contribute to AIC syndrome, although they are linked to other syndrome in database.  KMT2B and SMARCB1 play roles in epigenetic regulation of gene expression, which makes sense they are important. 

3.Authors introduced WNT8B and p.Leu70Pro variant and showed low protein stability in mutants, which is interesting, but how could link it to individual 1's AIC phenotype? 

4. For figure 2, still how to link wnt8, slf1 and szt2 phenotype to ACI?

Author Response

We thank the reviewer for their comments, please find our responses below.

  1. Could authors discuss more about: why AIC has a strong heterogeneous feature but only 4000 individuals are affected with AIC?

The combination of cortical malformations, infantile spasms and specific eye developmental defects which are required for AIC diagnosis are rare, even with the allowable heterogeneity of the diagnostic criteria, thus the numbers of individuals diagnosed remains low.  The following cited references define diagnostic criteria and prevalence.

Kroner, B.L.; Preiss, L.R.; Ardini, M.-A.; Gaillard, W.D. New Incidence, Prevalence, and Survival of Aicardi Syndrome from 408 Cases. J. Child Neurol. 2008, 23, 531–535, doi:10.1177/0883073807309782.

Aicardi, J. Aicardi Syndrome. Brain Dev. 2005, 27, 164–171, doi:10.1016/j.braindev.2003.11.011.

  1. AIC is diagnosed as a multiple mixed phenotypes, including agenesis of the corpus callosum, epileptic spasms and so on, could KMT2B and SMARCB1 also contribute to AIC syndrome, although they are linked to other syndrome in database. KMT2B and SMARCB1 play roles in epigenetic regulation of gene expression, which makes sense they are important.

Both SMARCB1 and KMT2B have distinct epigenetic signatures as identified by Aref-Eshghi et al. 2020. In the case of the SMARCB1 the variant we identified is a known cause of Coffin-Siris Syndrome and already noted in ClinVar, thus the initial diagnosis of AIC given at an early age was a misdiagnosis.  Inclusion of this patient demonstrates that the heterogeneity of traits associated with AIC can lead to inclusion of individuals with related disorders as part of the differential diagnosis.  In the case of KMT2B, more than 100 individuals have been described in whom focal lower-limb dystonia is a distinguishing trait. Dystonia was not observed in Individual 9 in our study thus given the lack of this pathognomonic feature the NM_014727.1 (KMT2B): c.6418C>G: p.(Pro2140Ala) variant is unlikely to be disease causing.

We have amended the Results section to highlight the frequency of dystonia in association with KMT2B variants.

Aref-Eshghi, E.; Kerkhof, J.; Pedro, V.P.; Groupe DI France; Barat-Houari, M.; Ruiz-Pallares, N.; Andrau, J.-C.; Lacombe, D.; Van-Gils, J.; Fergelot, P.; et al. Evaluation of DNA Methylation Episignatures for Diagnosis and Phenotype Correlations in 42 Mendelian Neurodevelopmental Disorders. Am J Hum Genet 2020, 106, 356–370, doi:10.1016/j.ajhg.2020.01.019.

  1. Authors introduced WNT8B and p.Leu70Pro variant and showed low protein stability in mutants, which is interesting, but how could link it to individual 1's AIC phenotype?

We suggest that WNT8B is an excellent candidate to explain AIC.  Overall our data support that the WNT8B p.Leu70Pro variant has a dominant negative effect on canonical WNT signalling (Figure 1a) even when the protein is destabilised (Figure 1a and 1b). In Figure 2a and 2c we show that loss of wnt8b causes eye and brain malformations suggestive of Aicardi syndrome.  Further work, beyond the scope of this study, such as a knock in mouse model is needed to prove that WNT8B p.Leu70Pro can cause AIC.

  1. For figure 2, still how to link wnt8, slf1 and szt2 phenotype to AIC?

The morphant phenotypes particularly the loss of eye pigmentation and developmental defects observed for slf1 and wnt8b, link these gene to the two target tissues in which AIC pathologies are observed.  These observations which mirror the effects of the published AIC gene tead1 promote these genes as plausible candidates for AIC.